# Up on the roof and down in the dirt: Differences in substrate properties (SOM, potassium, phosphorus and pH) and their relationships to each other between sedum and wildflower green roofs

Renée McAlister[ID]*, Anja S. Rott

Ecology, Conservation and Zoonosis Research and Enterprise Group, School of Pharmacy and Biomolecular Sciences, University of Brighton, Brighton, United Kingdom

☯ These authors contributed equally to this work.
* reneemcalister@gmail.com

**Data Availability Statement:** All relevant data are withing the manuscript and/or Supporting Information files.

## Abstract

In urban areas green roofs provide important environmental advantages in regard to biodiversity, storm water runoff, pollution mitigation and the reduction of the urban heat island effect. There is a paucity of literature comparing different types of green roof substrates and their contributions to ecosystem services or their negative effects. This study investigated if there was a difference between sedum and wildflower green roof substrate properties (soil organic matter (SOM), potassium (K) and phosphorus (P) concentrations and pH values) of 12 green roofs in the city of Brighton & Hove. One hundred substrate samples were collected (50 from sedum roof substrates and 50 from wildflower roof substrates) and substrate properties were investigated using standard protocols. Comparisons were made between substrate characteristics on both types of roof substrate with a series of multiple linear regressions. Sedum roofs displayed significantly higher values of SOM, P and pH. There were significant positive relationships between SOM and K concentrations, SOM and P concentrations, pH and K concentrations and pH and P concentrations on sedum roofs. This study concluded that sedum roof substrates are more favourable for plant water use efficiency and also contained a significantly higher percentage of SOM than wildflower roofs. However, higher concentrations of P in sedum roof substrates may have implications in regard to leachates.

## Introduction

Green roofs have been posited as a technology that can improve the urban environment in regard to biodiversity, storm water runoff, pollution mitigation and the reduction of the urban heat island effect. Research into green roofs is relatively recent [1] with only a small proportion of studies focussing on the contribution of green roofs to urban biodiversity [1–3] as the

**Funding:** This work was completed, in part, as an MSci research project by R McAlister, supervised by AS. Rott. Consumables were provided by the Biology Division, School of Pharmacy and Biomolecular Sciences, as part of the PG research project support. Identification of the surveyed roofs was through personal communication with Lee Evans, Organic Roofs Ltd.

**Competing interests:** The authors have declared that no competing interests exist.

predominant focus is on their mitigating capabilities. These include decreasing the urban heat island effect [4, 5], pollution abatement [6, 7], storm water retention and flood risk prevention [8–11] as well as reduction in energy bills [12, 13].

Green roof substrates are developed to take into consideration the weight bearing capacity of the roof, its water holding capacity and its ability to diffuse oxygen to plant roots [14]. They need to be lightweight, porous and free draining in order to provide the vegetation with an optimal growing environment whilst addressing the constraints of architecture and stressors. There is a scarcity of research pertaining to green roof vegetation and substrate [15] but some recent studies have investigated certain variables. For example, Bates et al, 2013 [16] examined the interactions between drought and substrate depth and Madre et al, 2013 [2] explored wildflower colonisation on roofs in relation to succession, substrate depth and roof height. Additionally, Gabrych et al, 2016 [15] conducted a study in Finland in regard to substrate depth and roof age.

The complexity and variety of green roof substrates presents the ecologist with a vast number of questions in regard to which substrate qualities are interacting with which. The majority of studies into green roof substrates have investigated substrate depth or substrate composition and their effect on vegetation, storm water attenuation and nutrient leaching [17–21]. For example a recent study examined the role of green roof substrates in the prevention of nutrient leaching by comparing different compositions in a laboratory environment and suggested that nitrogen (N) and phosphorus (P) leaching were at their highest soon after installation [22].

Green roof soil organic matter (SOM) concentrations have not been explored but the predominant ecosystem services provided by SOM are sustaining the biodiversity of plant life and carbon sequestration [23]. It has been suggested by van Groenigen et al, 2017 [24] that soils could potentially compensate for the increases in atmospheric $CO_2$ due to climate change but this is dependent on the characteristics of SOM, explicitly the amount of soil organic carbon (SOC). SOM has been seen to increase SOC in many studies as it has been seen to positively correlate with soil carbon storage [25–30].

Continued research into the substrate properties of green roofs may enhance this knowledge and elucidate in which ways green roofs are important in improving urban environments. This current study was a snap shot of 12 green roofs in Brighton and Hove and aimed to expand knowledge of green roof substrates on sedum and wildflower roofs in regard to substrate characteristics. Sedum roofs are planted with succulents and typically have a shallow substrate (2-12cm) and wildflower roofs consist of forbs native to the specific region and have a deeper substrate (11-20cm) [15]. Specifically this study assessed certain substrate properties (SOM, K, P concentrations & pH levels) in both roof types in order to inform better practice for continued green roof development.

## Methodology

### Study area

The data were gathered in August 2017 on 12 green roofs in Brighton & Hove (see Table 1) and the temperature on sampling days ranged from 18 ˚–23 ˚C. The average yearly temperatures in the city range from 1 ˚–25 ˚C with an average of 14 ˚C (Met Office, 2016). The city of Brighton & Hove is located at 50˚50'35"N, 0˚07'53"E covering an area of 87.5 km². The roofs surveyed comprised of wildflower roofs and sedum roofs (see Table 1 for details) with an aggregate and green compost based substrate. They were located through a local company (Organic Roofs Ltd, Brighton) and the University of Brighton, which has a number of green roofs located throughout its campuses. All building owners were consulted and permission was obtained from them to removal substrate samples for the purposes of the study.

**Table 1. Specifications and properties for green roofs surveyed in Brighton & Hove 2017.**

| Roof location | Establishment date | Roof area (m$^2$) | Number of samples | Roof type | Substrate type |
|---|---|---|---|---|---|
| **Varley Hub** [1]<br>50˚51'47.58"N, 0˚06'27.44"W | 2012 | 64 | 9 | Sedum | No information |
| **Huxley Building** [1]<br>50˚50'45.42"N, 0˚07'07.86"W | 2014 | 370 | 15 | Sedum | Sedum blanket (includes recycled brick)* |
| **Falmer Sport's Hall 1** [1]<br>50˚51'35.72"N, 0˚05'17.87"W | 2012 | 202 | 14 | Sedum | Sedum blanket (includes recycled brick)* |
| **Falmer Sport's Hall 2** [1]<br>50˚51'35.72"N, 0˚05'17.87"W | 2012 | 94 | 12 | Sedum | Sedum blanket (includes recycled brick)* |
| **Checkland Roofs (x 6)** [1]<br>50˚51'36.43"N, 0˚05'12.40"W | 2010 | 42m$^2$ per roof | 7 per roof | Wildflower | 30-90mm brick based with clay & compost |
| **Organic Roofs Ltd** [2]<br>50˚49'47.72"N, 0˚12'41.04"W | 2015 | 10 | 4 | Wildflower | Shire Ultralight** |
| **Richardson's Yard** [3]<br>50˚50'00.01"N, 0˚08'27.59"W | 2013 | 10 | 4 | Wildflower | Shire Ultralight** |

Ownership: [1] University of Brighton; [2] Organic Roofs Ltd; [3] Brighton Housing Trust

* Traditionally includes porous aggregate and composted green waste

** Made from secondary aggregates (aircrete and clay) and green waste compost (pH 8.5)

Includes the number of substrate samples taken on each roof. (see S2 Table for full lists of plant species per roof).

## Substrate surveys

A stratified random method was utilised to ensure representative sampling. Over the 12 roofs 100 substrate samples were taken, 50 for sedum roofs and 50 for wildflower roofs. The number of samples per roof was established using the method of Gabrych et al, 2016 [15] which was based on roof size (S1 Table). Roof size was measured using Google Earth ruler [31] and verified in the field with a tape measure.

A typical sample was approximately the size of a handful as suggested by Ward (2017, personal communication) and each sample was placed into a sealable plastic bag ensuring most of the air was removed. The substrate samples were then frozen for 3 days and, after freezing, placed in an oven in 250 ml beakers at 40 ˚C for 5 days to dry [32]. This is preferable to air drying as the shorter time necessary reduces the risk of microbial activity changing the status of the soil, and also lessens costs in relation to laboratory activity [33].

## Soil organic matter (SOM)

To measure soil organic matter the loss on ignition technique was used. Samples were placed into 50 ml beakers after weighing the empty beaker and taring the scale. The samples and beaker were then weighed again. Samples were then placed in a muffle furnace at 400 ˚C for 24 hours as suggested by [34] for soils containing compost. After removing from the muffle furnace the resulting sample mass was then remeasured to determine the amount of organic matter that had been lost in the process and this was calculated as the percentage of organic matter in the substrate.

## Substrate preparation

Each sample was then disaggregated in a pestle and mortar and half of each sample was retained in the event of future necessity. The substrate was then dry sieved through a 2 mm flat sieve and the homogenised fragments were placed in plastic bags and labelled. The percentages of fraction <2 mm and >2 mm were recorded. Sieving is predominantly used for particle size

analysis to determine soil structure [35], but here it was used to prepare the substrate for later analysis techniques.

## Potassium and phosphorus (K & P)

K and P measurements were taken using a PerkinElmer Inductively Coupled Plasma Optical Emission Spectrometer (ICP-OES). This method has been suggested to be a quick and accurate technique to determine K and P measurements [36, 37]. Samples <2 mm were prepared in a ratio of 0.1:3:7, substrate sample to nitric acid (HNO3) to deionised water. HNO3 (3 ml) was added to the substrate sample (0.1 g) in a fume cupboard and left to stand for 3 hours before adding 7 ml of deionised water. The analytes were then centrifuged and processed in the ICP-OES. Data for both K and P samples were recorded as mg/L.

## Statistical analysis

The substrate characteristics (% SOM, K mg/L, P mg/L and pH) were compared between types of roof substrate samples using Mann-Whitney tests as the data were non-normally distributed. Two outliers were removed from the wildflower % SOM data as they were skewing the results and did not represent the general trend according to Cook's distance [38, 39]. A principal components analysis (PCA) was used for data reduction purposes for relationships between substrate characteristics. Once PCA results were analysed a series of multiple linear regressions were performed to ascertain the relationships between substrate characteristics and their differences between sedum and wildflower roofs. Spearman's rank correlation coefficients and their p values were recorded for each regression analysis.

# Results

Substrate characteristics were compared for both sedum and wildflower roof types (Table 2) showing that sedum roofs contained a significantly higher amounts of SOM and P and higher pH levels. Sedum roofs had an average of 21% SOM and whilst wildflower roofs had an average of 9.6%. K levels did not significantly vary between the types of roof.

## Relationships between substrate characteristics

An initial exploration of the data using principal components analysis (PCA) was conducted for substrate characteristics. An initial exploration of the data using principal components analysis (PCA) was conducted for substrate characteristics. Percentage organic matter and nutrient loading accounted for the key proportion of variation in the two first components (89.6%). The nutrient variations of the roof substrates were positive correlated for phosphorus and negative for potassium. Substrate characteristics were then compared against each other and compared between both sedum and wildflower roof substrates. Percentage SOM and P concentrations and SOM and K concentrations were highly correlated on sedum roofs

**Table 2. Mean differences (± SD) between substrate characteristics in relation to sedum and wildflower green roof type.**

| Substrate characteristic | Sedum | Wildflower | Mann-Whitney W-score | P value |
|---|---|---|---|---|
| Soil organic matter (%) | 21 ± 11.86 | 9.6 ± 8.05 | 3303.0 | <0.0001 |
| Phosphorus (mg/L) | 0.1383 ± 0.0852 | 0.0548 ± 0.0206 | 3120.0 | <0.0001 |
| pH | 8.78 ± 0.15 | 8.32 ± 0.1 | 3754.5 | <0.0001 |
| Potassium (mg/L) | 0.1864 ± 0.0945 | 0.1761 ± 0.0597 | 2581.0 | N/S |

Results are shown for differences between sedum and wildflower roofs (n = 50 per roof type).

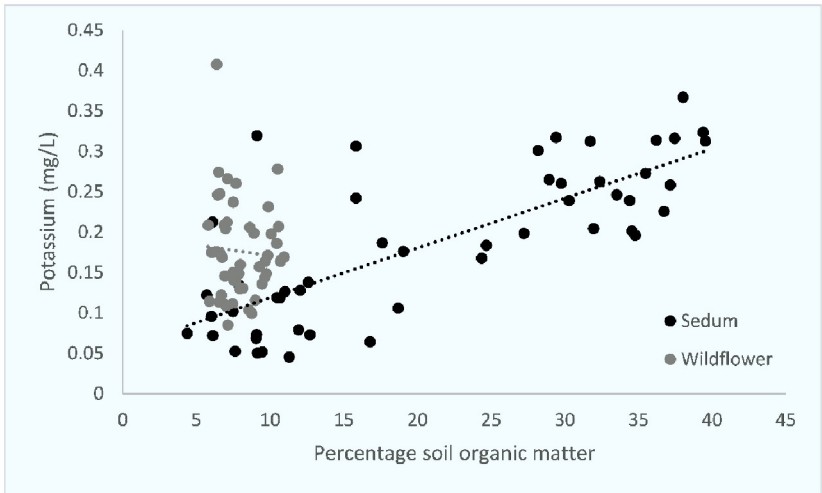

**Fig 1. The relationship between percentage soil organic matter and potassium on both sedum and wildflower roofs.** Two outliers have been removed from the wildflower % SOM data (42.42% and 52.9%). Sedum df = 49, r = 0.724, p = <0.0001. Wildflower df = 47, r = -0.020, p = N.S.

(Figs 1 and 2). pH and K concentrations and pH and P concentrations were also highly correlated on sedum roofs (Figs 3 and 4).

## Discussion

In the first study of its kind this research has suggested that substrate characteristics significantly differ between sedum and wildflower green roof substrates. SOM, phosphorus and pH values were significantly higher on sedum roofs. The relationships between substrate properties also differed significantly between sedum and wildflower green roofs, with increasing SOM correlated to higher potassium and phosphorus concentrations on sedum roofs and

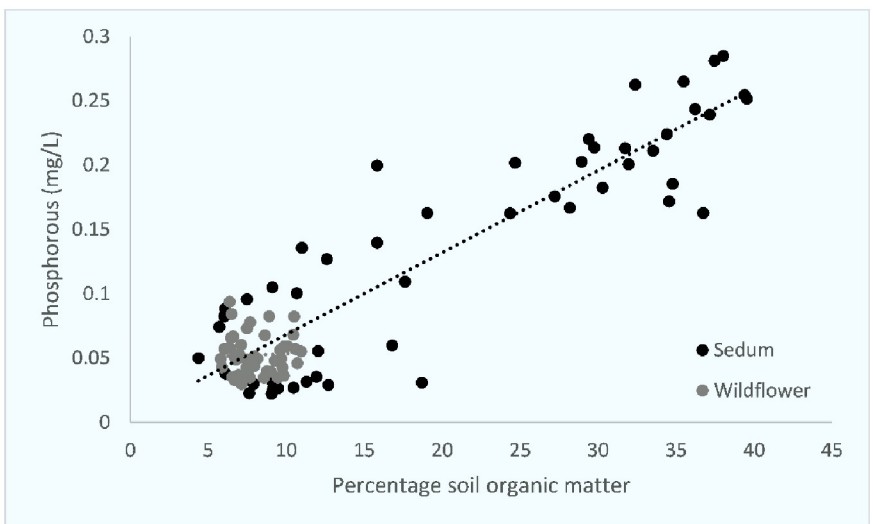

**Fig 2. The relationship between percentage soil organic matter and phosphorus on both sedum and wildflower roofs.** Two outliers have been removed from % SOM in wildflower roofs (42.42% and 52.9%). Sedum df = 49, r = 0.843, p = <0.0001. Wildflower df = 47, r = -0.079, p = N.S.

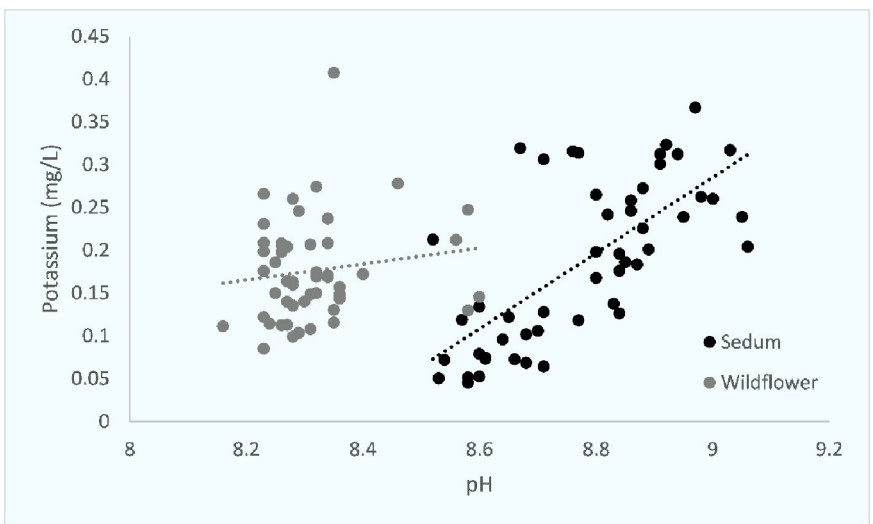

**Fig 3. The relationship between pH and potassium on both sedum and wildflower roofs.** Sedum df = 49, r = 0.696, p = <0.0001. Wildflower df = 49, r = 0.109, p = N.S.

increasing pH correlated to higher potassium and phosphorus concentrations on sedum roofs. However, given that it was not possible to determine the original substrate characteristics of the roofs and, therefore compare them, results are presented for consideration as a snapshot of substrate characteristics at the time of study.

## Soil organic matter

Human activities and industrialisation have been shown to have a negative impact on SOM in urban soils [40] hence any mitigation could have a strong positive impact in regard to carbon sequestration in cities. In this study sedum roof substrates displayed a significantly higher

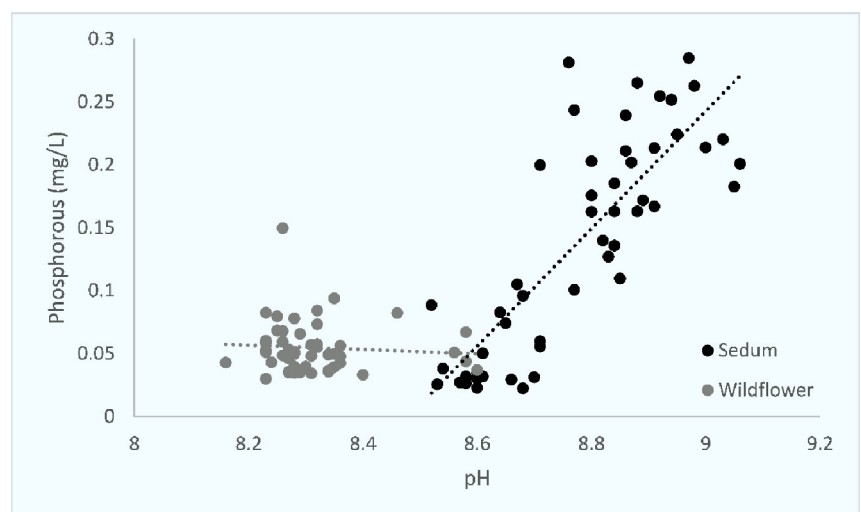

**Fig 4. The relationship between pH and phosphorus on both sedum and wildflower roofs.** Sedum df = 49, r = 0.814, p = <0.0001. Wildflower df = 49, r = 0.148, p = N.S.

SOM and, therefore, higher amount of SOC than wildflower roofs [25, 41] although, as substrates had different compositions, comparisons with initial substrates' SOM percentages could be of value. Green roofs in general have been shown to sequester carbon [42], however, as roofs age there are inconsistent results in regard to SOM levels as some studies reported increasing levels over time [9] while others have reported decreases [43]. Ground level urban green habitats have been shown to sequester carbon at higher levels [44]. The majority of carbon in cities is sequestered by urban trees and Xu et al, 2018 [45] suggested that, with their loss and the loss of other urban green space, it can take time for the system to recover, so with more green spaces encouraging higher biodiversity and connectivity in a fragmented landscape the more resilient it will be to disturbance. Urban environments also generate 'black carbon' from vehicular emissions [46] indicating that increasing carbon sinks (such as with suitable green roofs) will be necessary for climate change remediation considering that black carbon is a major contributor to climate change [47]. As urban areas continue to grow, with their concomitant reduction in ground level green space, green roofs may offer a lifeline in regard to urban carbon sequestration. This study observed a higher amount of SOM on sedum roofs, highlighting the importance of ensuring that green roof substrates address factors such as optimisation of plant growth and percentage of SOC in the green roof substrate.

Simulations have been conducted in relation to the mitigating effects of green roofs on climate change. It has been predicted that the temperature in the United States would rise 1–2 ˚C by 2100 if no mitigation strategies were adopted in urban areas and suggested green roofs as one of a number of adaptations to attenuate urban warming [48]. Additionally, Alcazar et al, 2016 [49] indicated that green roof systems have the potential to positively impact upon climate change in urban environments. Therefore, studies on the impact of green roof substrates on urban carbon sequestration are imperative as the consequences of climate change demand an urgent response and mitigation strategies may be a component of the solution.

## Phosphorus

Phosphorus levels from sedum roof substrate samples in this study were found to be significantly higher than the wildflower ones, which may have implications for nutrient runoff which can affect adjacent watercourse quality [50]. Wildflower roofs may have lower phosphorus concentrations than sedum roofs as they are regularly mowed, and mowing regimes have been seen to decrease phosphorus levels in arable grasslands [51].

Phosphorus can be stored in soils in inorganic and organic soil particles [52] posing the question as to whether green roof substrates are sources or sinks of phosphorus. Many green roof substrates are constructed to be nutrient rich and Mitchell et al, 2017 [53] connoted that green roof substrates can, in fact, be a source of phosphorus after initial installation up to a period of 10 or more years. However, they submit that phosphorus depletion may not be only due to runoff but could also be attributed to plant uptake and phosphorus chemical alteration in the substrate. The leaching of phosphorus from green roofs was seen to be significant in some studies [22, 54] which posited that younger green roofs had more phosphorus in their runoff. The initial composition of green roof substrate is paramount in the resultant leaching effects and adjacent habitat pollution as media type has a huge influence on whether green roofs are a source or sink of phosphorus [55]. It has been suggested that there are many factors involved in runoff dynamics on green roofs such as slope, soil moisture, rainfall and seasonal factors, age of the roof and vegetation [56]. However, the majority of studies have indicated that green roofs are generally sources of phosphorus pollution [56, 57] and this present study would suggest that it may be beneficial to investigate ways of mitigating phosphorus levels specifically in sedum roof substrates.

## pH

Both sedum and wildflower roofs in this study had relatively high pH values. *Sedum* species have been seen to thrive in a range of pH values [58] but no studies have suggested values as high as in this study. It has also been suggested that the average optimal pH for *Sedum* growth was 5.95, with non-optimal levels reducing growth dramatically [58]. Higher pH values have been shown to increase shoot growth in *Sedum* species during the winter period [59] which is undesirable as it puts extra stress on the plant.

A high amount of species on wildflower roofs were chalk grassland natives which thrive in soils with strong alkalinity. It has been seen that species richness decreases in these species with reduced pH levels [60]. However, Basto et al, 2015 [61] indicated that certain wildflower species' seed banks decrease with an increase in pH which has implications for the future species richness on wildflower roofs. It has been suggested that species richness in grasslands is reduced when mowing is stopped, which could indicate that the twice yearly mowing regime is important in regulating pH levels by reducing nutrients on wildflower green roofs [62].

### The relationships between substrate characteristics

There was a strong correlation between SOM and potassium on sedum roof substrates in this study. It has been shown that SOM adsorbs potassium at a quicker and much higher rate than mineral constituents [63] in the soil and potassium improves water use efficiency (WUE) in plants [64, 65]. Soil Organic Matter has also been seen to directly affect the dynamic processes that make potassium available to plants [66]. Increased SOM in soils increases the cation exchange capacity and potassium, being of low positive charge, which is then more easily taken up by plants [63]. As well as improving potassium exchangeability into available forms SOM assists potassium in reducing soil acidity [67], perhaps illustrating the high pH values in this study. However, in this study SOM and potassium in wildflower roofs had no significant relationship.

In this study there was also a significant strong relationship between SOM and total phosphorus on sedum roof substrates. The amount of SOM in soils has been seen to be correlated with higher available phosphorus concentrations [68–70]. However, negatively charged functional groups in SOM such as carboxyl groups can interact with iron oxides in the SOM which can increase phosphorus adsorption thus binding it to soil particles and making it unavailable to plants [71]. Many biogeochemical factors affect the availability of phosphorus in the soil such as soil moisture, SOM and clay content [72] as well as the interaction of humic acids with some metal oxides [73]. Although the relationship between SOM and phosphorus is a complex one, it is generally suggested that higher SOM fractions in soil increase phosphorus availability [74, 75]. As phosphorus is a limiting factor in plant growth the fact that sedum roofs had a high SOM versus phosphorus correlation in this study suggested that sedum roof substrates may be a healthier ecosystem for *Sedum* survivability.

Soil pH has an effect on both potassium and phosphorus availability and, in this study, sedum roofs were seen to have a strong to very strong positive relationship between the two variables. There is a paucity of literature in regard to these factors in soils but non-peer reviewed data sources have suggested that there is a relationship between pH and both potassium and phosphorus availability to vegetation. Hargreaves, 2015 [76] suggested that, in elevated levels of pH, the dominating ion is calcium (Ca) and the greater amount of Ca ions in the soil will increase the availability of potassium. As stated before, potassium increases the WUE of plants, suggesting that the higher pH on sedum roofs is beneficial in this respect. However, Hargreaves, 2015 [76] also indicated that increased Ca ions actually reduce the availability of phosphorus but this study found that, on sedum roofs, phosphorus concentrations

actually increased with increasing pH levels very significantly. This is an interesting result, as Westermann, 1992 [77] and Hopkins & Ellsworth, 2005 [78] also posited that alkaline soils impede the amount of available phosphorus due to the formation of calcium phosphate minerals which have poor solubility.

This present study investigated the presence of total phosphorus in green roof substrates rather than available (Olsen) phosphorus which may explain these results. Although the substrate on sedum roofs contained a significantly higher amount of total phosphorus than wildflower roofs in relation to pH, many alkaline soils have been seen to have relatively low amounts of available phosphorus in relation to total phosphorus [79]. Some studies have investigated solutions to this problem and have examined various soil additives which may decrease pH in order to increase phosphorus availability. For example, Liu et al, 2013 [80] advanced the addition of potassium sulfate ($K_2SO_4$) to substrates to reduce pH levels and microbial interactions within the soil have been seen to increase the amount of available phosphorus due to their ability to solubilise phosphorus [81].

## Conclusions

This study indicated that there is a difference between the majority of substrate properties on sedum and wildflower roofs, highlighting the need to consider the complex associations between green roof variables in order to further scientific knowledge and inform industry. Results suggested that sedum roofs provide more desirable roof substrate properties than wildflower roofs for certain ecosystem services such as soil and water conservation. The key findings demonstrated significantly higher SOM amount on sedum roofs compared to wildflower roofs. Soils higher in SOM are also higher in SOC and so the amount of carbon stored may be higher in sedum roofs. However, in this study sedum and wildflower roofs were installed with varying initial substrates and these may well have had an influence on the differences in SOM percentages. Sedum roofs also displayed higher concentrations of phosphorus which can lead to nutrient runoff that can negatively affect both urban and adjacent aquatic habitats, specifically in the earlier stages of the green roof life.

There have been no studies collecting data in regard to relationships between substrate properties on green roofs and this study represents a fresh approach in examining these variables. The positive relationship between SOM and potassium on sedum roofs was significant and since SOM can increase the availability of potassium in soils, the results indicate that the increased SOM in sedum roofs aids vegetation in regard to water use efficiency (WUE). Soil Organic Matter can also increase available phosphorus in soils and, considering that phosphorus is a limiting factor in plant growth, this relationship would seem to be beneficial in regard to *Sedum* health on green roofs. However, dependent on the level of phosphorus values there are issues in regard to phosphorus leachates. The higher pH on sedum roofs would suggest that there would be a higher amount of available potassium for these species, again increasing WUE and survivability of *Sedum* species. There are many variables to take into consideration when deciding whether to install sedum or wildflower green roofs and this study concludes that green roof choice can be dependent on which ecosystem services are desired whilst also taking into consideration green roof runoff.

## Supporting information

**S1 Table. Number of samples based on roof size.**
(DOCX)

**S2 Table. Plant species lists for individual roofs.**
(DOCX)

## Acknowledgments

We would like to thank Organic Roofs Ltd., Brighton for providing essential information in regards to the specifications of the green roofs. Also, thank you to Brighton Housing Trust and the University of Brighton for access to the roofs. Additionally, thanks to Will Mills, Magdalena Grove and Pete Lyons for their knowledge and support.

## Author Contributions

**Conceptualization:** Anja S. Rott.

**Data curation:** Renée McAlister.

**Formal analysis:** Renée McAlister.

**Investigation:** Renée McAlister.

**Methodology:** Renée McAlister.

**Project administration:** Anja S. Rott.

**Supervision:** Anja S. Rott.

**Writing – original draft:** Renée McAlister.

**Writing – review & editing:** Anja S. Rott.

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
