## [Editor Report · Decision Letter 0]

18 Jul 2019

PONE-D-19-19569

Up on the roof and down in the dirt: differences in soil properties (SOM, potassium, phosphorus and pH) and their relationship to each other between sedum and wildflower green roofs.

PLOS ONE

Dear Ms McAlister,

Thank you for submitting your manuscript to PLOS ONE. After careful consideration, we feel that it has merit but does not fully meet PLOS ONE’s publication criteria as it currently stands. Therefore, we invite you to submit a revised version of the manuscript that addresses the points raised during the review process.

ACADEMIC EDITOR: Please address the comments listed under "Additional Editor Comments" 

We would appreciate receiving your revised manuscript by 30 July 2019. To enhance the reproducibility of your results, we recommend that if applicable you deposit your laboratory protocols in protocols.io, where a protocol can be assigned its own identifier (DOI) such that it can be cited independently in the future. For instructions see: http://journals.plos.org/plosone/s/submission-guidelines#loc-laboratory-protocols

We look forward to receiving your revised manuscript.

Kind regards,

Luitgard Schwendenmann

Academic Editor

PLOS ONE

Journal Requirements:

3. In your Methods, please include a specific statement that permission was obtained from all building owners for the removal of green roof soil samples for the purposes of your study.

Additional Editor Comments:

While the topic is of interest, I would like to ask for further details regarding the methods and a critical assessment of the findings. 7 roofs were selected and a varying number of samples (depending on the size of the roof) were sampled. What is the reasoning to consider each sample taken at a given roof as independent (n=50). In my opionion these are pseudo-replicates. Further, did you test for an age effect (time since establishment) and even more critical is the difference between the substrates. It remains unclear whether your questions relate to substrate (e.g. line 163) or plant type (e.g. 172/173). How can substrate and plant type be separated (what is cause and effect)? Another aspect which needs to be addressed before I consider sending the manuscript for review is the difference between the carbon content at the time of establishment (baseline - carbon emboddied in the substrate) and time of sampling. In my opinion it is more relevant to consider the difference (not the total amount). I am looking forward to your response.
---

## [Author Response · Author response to Decision Letter 0]

1 Aug 2019

I am very thankful for your kind comments on the manuscript following the submission for publication. 

The manuscript has been revised in line with your comments (formatting and lines 112, 171 and 747), and I am happy to respond to your remarks below:

7 roofs were selected and a varying number of samples (depending on the size of the roof) were sampled. What is the reasoning to consider each sample taken at a given roof as independent (n=50). In my opinion these are pseudo-replicates.

A total of 12 roofs were surveyed, as the Checkland building had 6 individual green roofs of different heights and aspects. I was guided by the methodology of green roof researchers Gabrych M., Kohtze, D.J & Lehvävirta, 2016. Substrate depth and roof age affect plant abundance on sedum-moss and meadow green roofs in Helsinki, Finland. They published their work using 475 sample plots over 51 roofs, basing the sample size per roof on roof size. Ecological in situ studies are naturally imposed with limitations but our results indicate there was soil property variability not just between roofs, but within roofs. This was a site-specific snap shot of the soil properties of 12 green roofs in Brighton and Hove and, although its pseudo-replication has been questioned, I believe it has scientific merit as this is one of the first studies to analyse these particular four properties in green roof soils. 

Further, did you test for an age effect (time since establishment) and even more critical is the difference between the substrates. It remains unclear whether your questions relate to substrate (e.g. line 163) or plant type (e.g. 172/173).

The ages of the roofs were recorded but not included in the statistical analysis. As this was a snap shot of the present situation on the green roofs, we did not incorporate this data. The different types of substrates were not analysed independently, only the soil samples (lines 95 – 98).). The question in regard to substrate/plant type has been addressed in line 171 which, we believe, clarifies your enquiry. The questions relate to the difference in soil properties between the two green roof types, sedum and wildflower 

Another aspect which needs to be addressed before I consider sending the manuscript for review is the difference between the carbon content at the time of establishment (baseline - carbon embodied in the substrate) and time of sampling. In my opinion it is more relevant to consider the difference (not the total amount).

I was not able to measure the carbon baseline of the roofs at installation and nor were the owners of the roofs able to provide us with this information. Therefore this study was a scientific depiction of the present state of the roofs, rather than a comparison between their soil properties at installation and at the time of surveying. 

Once more, thank you for your comments and the opportunity to respond. I look forward to hearing from you.

Sincerely

Renée McAlister

---

## [Decision Letter · Decision Letter 1]

23 Sep 2019

PONE-D-19-19569R1

Up on the roof and down in the dirt: differences in soil properties (SOM, potassium, phosphorus and pH) and their relationship to each other between sedum and wildflower green roofs.

PLOS ONE

Dear Ms McAlister,

Thank you for submitting your manuscript to PLOS ONE. After careful consideration, we feel that it has merit but does not fully meet PLOS ONE’s publication criteria as it currently stands. Therefore, we invite you to submit a revised version of the manuscript that addresses the points raised during the review process.

This paper provides a good snapshot but further revisions are required.

I agree with reviewer 1. If the material is an engineered green roof it is not appropriate to refer to them as soil. I recommend to use the term "substrate" throughout the manuscript. And replace "edaphic" by substrate characteristics.

The authors have to address this concern ("However, I don’t know how any conclusions can be made that sedum may be better than wildflowers at sequestering carbon. Are the differences in all the variables due to the plant type or to the differences in the soil components? The soils on the sedum roofs were composed of recycled bricks while the soils on the wildflower roofs were composed of an assortment of different aggregates and composts. Differences in SOM, pH, and P likely existed before plants were even installed. Maybe plant type had no effect.") and provide further clarification. This is a critical limitation of the study and needs to be discussed.

We would appreciate receiving your revised manuscript by Nov 07 2019 11:59PM. To enhance the reproducibility of your results, we recommend that if applicable you deposit your laboratory protocols in protocols.io, where a protocol can be assigned its own identifier (DOI) such that it can be cited independently in the future. For instructions see: http://journals.plos.org/plosone/s/submission-guidelines#loc-laboratory-protocols

We look forward to receiving your revised manuscript.

Kind regards,

Luitgard Schwendenmann

Academic Editor

PLOS ONE

Additional Editor Comments (if provided):

This paper provides a good snapshot but further revisions are required.

I agree with reviewer 1. If the material is an engineered green roof it is not appropriate to refer to them as soil. I recommend to use the term "substrate" throughout the manuscript. And replace "edaphic" by substrate characteristics.

The authors have to address this concern ("However, I don’t know how any conclusions can be made that sedum may be better than wildflowers at sequestering carbon. Are the differences in all the variables due to the plant type or to the differences in the soil components? The soils on the sedum roofs were composed of recycled bricks while the soils on the wildflower roofs were composed of an assortment of different aggregates and composts. Differences in SOM, pH, and P likely existed before plants were even installed. Maybe plant type had no effect.") and provide further clarification. This is a critical limitation of the study and needs to be discussed.

Reviewers' comments:

Reviewer's Responses to Questions

**Comments to the Author**

1. If the authors have adequately addressed your comments raised in a previous round of review and you feel that this manuscript is now acceptable for publication, you may indicate that here to bypass the “Comments to the Author” section, enter your conflict of interest statement in the “Confidential to Editor” section, and submit your "Accept" recommendation.

Reviewer #1: (No Response)

Reviewer #2: (No Response)

2. Is the manuscript technically sound, and do the data support the conclusions?

Reviewer #1: Yes

Reviewer #2: Partly

3. Has the statistical analysis been performed appropriately and rigorously? 

Reviewer #1: Yes

Reviewer #2: I Don't Know

4. Have the authors made all data underlying the findings in their manuscript fully available?

Reviewer #1: Yes

Reviewer #2: Yes

5. Is the manuscript presented in an intelligible fashion and written in standard English?

Reviewer #1: Yes

Reviewer #2: Yes

6. Review Comments to the Author

Reviewer #1: I noticed that you have changed substrate to soil in many places in the manuscript (eg lines 96 , 171 in the revised manuscript with track changes). If the substrates on the green roofs really are engineer green roof media, I am not sure it is appropriate to refer to them as soils. I do understand that this can make the terminology cumbersome, but I think it is an important distinction.

Introduction

Lines 49-51: the list is missing commas.

Results

Lines 197-199: you say the first two components account for 89.6% of variation, which components are these?

Discussion:

Line 293: “However, the caveat that phosphorus…” this wording is a bit odd. I would suggest replacing caveat with another word.

Line 376: “connoting” seems like the wrong word to use hear.

Reviewer #2: The paper provides a good snapshot in time of the soil properties present at the time of measurement. However, I don’t know how any conclusions can be made that sedum may be better than wildflowers at sequestering carbon. Are the differences in all the variables due to the plant type or to the differences in the soil components? The soils on the sedum roofs were composed of recycled bricks while the soils on the wildflower roofs were composed of an assortment of different aggregates and composts. Differences in SOM, pH, and P likely existed before plants were even installed. Maybe plant type had no effect. There needs to be a soil analysis of the original soils. This is impossible now, but the soil samples could be mixed with the original ingredients that would give a reasonable estimate of the baseline properties.

Also, there is little information on plants. What is a sedum roof and what is a wildflower roof? Any grasses? Plant species could have a major impact on soil properties. A list of plant species should be included to describe each roof.

7. PLOS authors have the option to publish the peer review history of their article (what does this mean?). If published, this will include your full peer review and any attached files.

Reviewer #1: No

Reviewer #2: No

---

## [Author Response · Author response to Decision Letter 1]

7 Nov 2019

Associate Professor Luitgard Schwendenmann

Academic Editor – PLOS ONE

Dear Professor Schwendenmann

I am very thankful for the kind comments of your reviewers on the manuscript following my initial revisions. 

The manuscript has been revised in line with their comments. All references to soil have been changed to substrate, the missing commas (lines 52-53) have been added and the first two components of the PCA (lines 209-216) have been explained. Additionally, the wording has been changed on lines 320 and 400 in accordance with their suggestions (line 320 caveat has been changed to submit and line 404 connoting has been changed to suggesting).

However, I don’t know how any conclusions can be made that sedum may be better than wildflowers at sequestering carbon. Are the differences in all the variables due to the plant type or to the differences in the soil components?

I agree with the reviewers in regard to my conclusions about carbon sequestration as these conclusions were rather over speculative. I have therefore removed all references to this inference. 

Also, there is little information on plants. What is a sedum roof and what is a wildflower roof? Any grasses? Plant species could have a major impact on soil properties. A list of plant species should be included to describe each roof.

I have included lists of all plant species on all roofs in Supporting Information 2 (signposted in Table 1). Additionally, I have added an explanatory sentence in the introduction (lines 97-100) giving a brief overview of the differences between sedum and wildflower green roofs. 

The soils on the sedum roofs were composed of recycled bricks while the soils on the wildflower roofs were composed of an assortment of different aggregates and composts. Differences in SOM, pH, and P likely existed before plants were even installed. Maybe plant type had no effect.

I believe I have addressed this concern in lines 256-260 by qualifying that the research is presented for consideration as a snapshot of the substrate characteristics at the time of study. I have also added information that comparing the initial substrates in regard to SOM would be beneficial (lines 270-272) and included information regarding opposing studies which report either increasing or decreasing levels of SOM over time on green roofs (lines 273-276).

Additionally, Thuring & Dunnett (2014) suggested that green roof substrates do change over time in regard to their characteristics and therefore I believe this snapshot is a good representation of the current substrate characteristics taking aging into consideration. The roofs I studied were between 2 and 7 years of age, and Thuring & Dunnett (2014) indicate that substrate characteristics can indeed alter in a 2 year time frame.

Once again, thank you very much for your comments and the opportunity to respond. I look forward to hearing from you.

Sincerely

Renée McAlister

Thuring, C.E. & Dunnett, N., 2014. Vegetation composition of old extensive green roofs (from 1980s Germany). Ecological Processes. 3 (4) pp.1-11.

---

## [Editor Report · Decision Letter 2]

11 Nov 2019

Up on the roof and down in the dirt: differences in substrate properties (SOM, potassium, phosphorus and pH) and their relationship to each other between sedum and wildflower green roofs.

PONE-D-19-19569R2

Dear Dr. McAlister,

We are pleased to inform you that your manuscript has been judged scientifically suitable for publication and will be formally accepted for publication once it complies with all outstanding technical requirements.

With kind regards,

Luitgard Schwendenmann

Academic Editor

PLOS ONE

Additional Editor Comments (optional):

Thank you for replacing "soil" by "substrate". I highly recommend to replace "soil organic matter" by "organic matter" for consistency.
---

## [Editor Report · Acceptance letter]

5 Dec 2019

PONE-D-19-19569R2 

Up on the roof and down in the dirt: differences in substrate properties (SOM, potassium, phosphorus and pH) and their relationships to each other between sedum and wildflower green roofs. 

Dear Dr. McAlister:

I am pleased to inform you that your manuscript has been deemed suitable for publication in PLOS ONE. Congratulations! Your manuscript is now with our production department. 

With kind regards,

on behalf of

Dr. Luitgard Schwendenmann 

Academic Editor

PLOS ONE